# RandAlign: A Parameter-Free Method for Regularizing Graph Convolutional Networks

## Abstract

Studies continually find that message-passing graph convolutional networks suffer from the over-smoothing issue. Basically, the issue of over-smoothing refers to the phenomenon that the learned embeddings for all nodes can become very similar to one another and therefore are uninformative after repeatedly applying message passing iterations. Intuitively, we can expect the generated embeddings become smooth asymptotically layerwisely, that is each layer of graph convolution generates a smoothed version of embeddings as compared to that generated by the previous layer. Based on this intuition, we propose RandAlign, a stochastic regularization method for graph convolutional networks. The idea of RandAlign is to randomly align the learned embedding for each node with that generated by the previous layer using random interpolation in each graph convolution layer. Through alignment, the smoothness of the generated embeddings is explicitly reduced. To better maintain the benefit yielded by the graph convolution, in the alignment step we introduce to first scale the embedding of the previous layer to the same norm as the generated embedding and then perform random interpolation for aligning the generated embedding. RndAlign is a parameter-free method and can be directly applied without introducing additional trainable weights or hyper-parameters. We experimentally evaluate RandAlign on different graph domain tasks on seven benchmark datasets. The experimental results show that RandAlign is a generic method that improves the generalization performance of various graph convolutional network models and also improves the numerical stability of optimization, advancing the state of the art performance for graph representation learning.

## 1 Introduction

Graph-structured data are very commonly seen in the real world (Hamilton, 2020; Feng et al., 2020). Social networks, protein and drug structures, 3D meshes and citation networks—all of these types of data can be represented using graphs. It is of considerable significance to design and develop models that are able to learn and generalize from this kind of data. The past years have seen a surge in studies on representation learning on graph-structured data, including techniques for deep graph embedding, graph causal inference and generalizations of convolutional neural networks to non-Euclidean data (Hamilton, 2020). These advances have produced new state of the art results in a wide variety of domains, including recommender systems, drug discovery, 2D and 3D computer vision, and question answering systems (Sun et al., 2020; Chen et al., 2022b; Guo et al., 2021; Huang et al., 2020; Deng et al., 2022).

Unlike images and natural languages, which essentially have a grid or sequence structure, graph-structured data have an underlying structure in non-Euclidean spaces. It is a complicated task to develop models that can generalize over general graphs. Early attempts (Gori et al., 2005; Scarselli et al., 2008) on graph representation learning primarily use a recursive neural network which iteratively updates node states and exchanges information until these node states reach a stable equilibrium. Recent years have seen the popularity of graph convolutional networks for graph-structured data. Graph convolutional networks are derived as a generalization of convolutions to non-Euclidean data (Bruna et al., 2014). The fundamental feature of graph convolutional networks is that it utilizes a message passing paradigm in which messages are exchanged between nodes and updated using neural networks (Gilmer et al., 2017).

This paradigm of message passing is basically a differentiable variant of belief propagation (Dai et al., 2016). During each message-passing iteration, the representation for each node is updated according to the information aggregated from the node's neighborhood. This local feature-aggregation behaviour is analogous to that of the convolutional

kernels in convolutional neural networks, which aggregates feature information from spatially-defined patches in an image. Message passing is the core of current graph convolutional networks, but it also has major drawbacks. Theoretically, the power of message-passing graph convolutional networks is inherently bounded by the Weisfeiler-Lehman isomorphism test (Xu et al., 2019; Morris et al., 2019). Empirically, studies have continually found that message-passing graph neural networks suffer from the problem of over-smoothing, and this issue of over-smoothing can be viewed as a consequence of the neighborhood aggregation operation (Hamilton, 2020).

The problem of over-smoothing is that after a number of message passing iterations, the representations for all the nodes in the graph can become very similar to one another. This is problematical because node-specific information becomes lost when we add more graph convolutional layers to the model. This makes it difficult to capture long-term dependencies in the graph using deeper layers. Due to the over-smoothing issue, basic graph convolutional network models such as GCN (Kipf & Welling, 2016) and GAT (Veličković et al., 2018) are restricted to a small number of layers, e.g., 2 to 4 (Zhao & Akoglu, 2020). Further increasing the number of layers will lead to significantly reduced generalization performance. This is different from convolutional neural networks, the performance of which can be considerably improved by using very deep layers. Study also shows that the issue of over-smoothing can cause overfitting or underfitting for different graph domain tasks (Zhang et al., 2022).

Increasing efforts have been devoted to understanding and addressing the over-smoothing problem over the past years. From the graph signal processing view, applying message passing in a basic graph convolutional network is analogous to applying a low-pass convolutional filter, which produces a smoothed version of the input signal (Zhu & Koniusz, 2020). Li et al. (Li et al., 2018) showed that graph convolution is a special form of Laplacian smoothing (Taubin, 1995) and proved that repeatedly applying Laplacian smoothing can lead to node representations becoming indistinguishable from each other. Zhao et al. (Zhao & Akoglu, 2020) proposed a normalization layer named PariNorm that ensures the total pairwise feature distance remains unchanged across layers to prevent node features from converging to similar values. Zhang et al. (Zhang et al., 2022) introduced to stochastically scale features and gradients (SSFG) during training. This method explicitly breaks the norms of generated embeddings becoming over-smoothed for alleviating over-smoothing.

As introduced above, the learned embeddings for all nodes become very similar to one another when over-smoothing occurs. When it comes to becoming very similar to one another, we can understand it from two respects: (1) These embeddings have a small cosine similarity between one and another; (2) The norms of these embeddings are close to each other. The SSFG method (Zhang et al., 2022) is effective through addressing the norms of node embeddings converging to the same value with regard to the second respect. However, the issue of node embeddings having a small cosine similarity between one and another is explicitly addressed by the SSFG method. As aforementioned, the over-smoothing problem comes after repeatedly message passing iterations. Intuitively, we can expect the learned embeddings for the nodes become smoothed layerwisely or asymptotically layerwisely. That is, each message passing iteration produces a smoothed version of the input embeddings. In this paper we first show, through an example, the intuition that each layer of graph convolution can make the generated node embeddings closer to each other than the input embeddings. Based on this intuition, we propose RandAlign, a stochastic regularization method for graph convolutional networks. The idea of RandAlign is to randomly align the generated embedding for each node with that generated by the previous layer. Because the embeddings generated by the previous layer are less close to each other, the problem of over-smoothing with regard to the first respect is explicitly reduced through alignment.

In alignment, we sample a factor from the standard uniform distribution and then align the generated embedding for each node with that generated by the previous layer using convex combination. Therefore our RandAlign method does not introduce additional trainable parameters or hyper-parameters. It can be applied to current message-passing graph convolutional networks in plug and play manner. We show through experiments that RandAlign is a generic method that improves the generalization performance of a variety of graph convolutional networks including GCN (Kipf & Welling, 2017), GAT (Veličković et al., 2018), GatedGCN (Bresson & Laurent, 2017), SAN (Kreuzer et al., 2021) and GPS (Rampasek et al., 2022). We also show that RandAlign is effective on seven popular datasets on different graph domain tasks, including graph classification and node classification, advancing the state of the art results for graph representation learning on these datasets.

The main contributions of this paper can be summarized as follows:

- We propose a stochastic regularization method named RandAlign for graph convolutional networks. RandAlign randomly aligns the learned embedding for each node with that learned by the previous layer using

random interpolation. This explicitly reduces the smoothness of the generated embeddings. Moreover, we introduce to first scale the embedding of the previous layer to the same norm as the generated embedding and then perform random interpolation for aligning the generated embedding. This scaling step helps to maintain the benefit yielded by graph convolution in the aligned embeddings.

- RandAlign is a parameter-free method which does not introduce additional trainable parameters or hyper-parameters. It can be directly applied to current graph convolutional networks without increasing the model complexity and the parameter tuning procedure.

- We demonstrate that RandAlign is a generic method that consistently improves the generalization performance of various graph convolutional network models, advancing the state of the art results on different graph domain tasks on seven popular benchmark datasets. We also show that RandAlign helps to improve the numerical stability of optimization.

## 2 Related Work

### 2.1 Graph Convolutional Networks

The first-generation graph neural work models were developed by Gori et al. (Gori et al., 2005) and Scarselli et al. (Scarselli et al., 2008). These models generalize recursive neural networks for general graph-structured data. Motivated by the success of convolutional neural networks for Euclidean data, recent years have seen increasing studies on graph convolutional networks which generalize Euclidean convolutions to the non-Euclidean graph domain. Current graph convolutional networks can be categorized into spectral approaches and spatial approaches (Wu et al., 2020).

The spectral approaches are based on spectral graph theory. The key idea in these approaches is that they construct graph convolutions via an extension of the Fourier transform to graphs, and a full model is defined by stacking multiple graph convolutional layers. For example, Bruna et al. (Bruna et al., 2014) proposed to construct graph convolutions based on the eigendecomposition of the graph Laplacian. Following on Bruna's work, Defferrard et al. (Defferrard et al., 2016) introduced to construct convolutions based on the Chebyshev expansion of the graph Laplacian. This approach eliminates the process for graph Laplacian decomposition and results in spatially localized filters. Kipf and Welling (Kipf & Welling, 2017) simplified the previous methods by introducing the popular GCN architecture, wherein the filters are defined on the 1-hop neighbourhood as well as the node itself.

Unlike the spectral approaches, the spatial approaches directly define convolutions on the graph and generate node embeddings nodes by aggregating information from a local neighbourhood. Monti et al. (Monti et al., 2017) proposed a mixture model network, referred to as MoNet, which is a spatial approach that generalizes convolutional neural network architectures to graphs and manifolds. Velickovic et al. (Veličković et al., 2018) introduced to integrate the self-attention mechanism which assigns an attention weight or importance value to each neighbour in local feature aggregation into graph convolutional network models. Bresson et al. (Bresson & Laurent, 2017) proposed residual gated graph convnets, integrating edge gates, residual connections (He et al., 2016) and batch normalization (Ioffe & Szegedy, 2015) into the graph convolutional neural network model. Balcilar et al. (Balcilar et al., 2021) demonstrated that both spectral and spatial graph convolutional networks are essentially message passing neural networks that use a form of message passing for node embedding generation.

### 2.2 The Over-smoothing Problem

Over-smoothing is a common issue with current graph convolutional neural networks. Intuitively, this phenomenon of over-smoothing occurs when the information aggregated from the local neighbours starts to dominate the updated node embeddings. Therefore, a straightforward way to reduce over-smoothing is to use feature concatenations or skip connections (Hamilton, 2020), which are commonly used in computer vision to build deep convolutional network architectures. Feature concatenations and skip connections can preserve information learned by previous graph convolutional layers. Inspired by the gating methods used to improve recurrent neural networks, researchers also proposed gated updates in aggregating information from local neighbours (Li et al., 2015; Bresson & Laurent, 2017). These gated updates are very effective in building deep graph convolutional network architectures, e.g., 10 or more layers. Zhao et al. (Zhao & Akoglu, 2020) proposed the PairNorm method to tackle oversmoothing by ensuring the total

pairwise feature distance across layers to be constant. Zhang et al. (Zhang et al., 2022) proposed a stochastical regularization method called SSFG that randomly scales features and gradients in the training procedure. Empirically, SSFG can help to address both the overfitting issue and the underfitting issue for different graph domain tasks. Chen et al. (Chen et al., 2022c) proposed a graph convolution operation, referred to as graph implicit nonlinear diffusion, that can implicitly have access to infinite hops of neighbours while adaptively aggregating features with nonlinear diffusion to alleviate the over-smoothing problem.

## 3 Methodology

In this section, we begin by introducing the notations and the message passing framework. Then we introduce the over-smoothing issue with graph convolutional networks. Finally we describe the proposed RandAlign method for regularizing graph convolutional networks through reducing the over-smoothing problem.

### 3.1 Preliminaries

Formally, a graph $G = (V, E)$ can be defined by a set of nodes, or called vertices, $V$ and a set of edges $E$ between these nodes. An edge going from node $u \in V$ to node $v \in V$ is denoted as $(u, v)$. Conveniently, the graph $G$ can be represented using an adjacent matrix $\mathbf{A} \in \mathbb{R}^{|V| \times |V|}$, in which $\mathbf{A}_{u,v} = 1$ if $(u, v) \in E$ or $\mathbf{A}_{u,v} = 0$ otherwise. The degree matrix of $G$ is a diagonal matrix and is denoted as $\mathbf{D} \in \mathbb{R}^{|V| \times |V|}$, in which $\mathbf{D}_{ii} = \sum_j \mathbf{A}_{ij}$. The node-level feature or attribute associated with $u \in V$ is denoted as $\mathbf{x}_u$. The graph Laplacian is defined as $\mathbf{L} = \mathbf{D} - \mathbf{A}$, and the symmetric normalized Laplacian is defined as $\mathbf{A}_{sym} = \mathbf{I}_n - \mathbf{D}^{-1/2} \mathbf{A} \mathbf{D}^{-1/2}$, where $\mathbf{I}_n$ is a $|V| \times |V|$ identity matrix.

Message passing is at the core of current graph convolutional networks. In the message passing paradigm, nodes aggregated message from neighbours and updated their embeddings according to the aggregated information in an iterative manner. This message passing update can be expressed as follows (Hamilton, 2020):

$$\mathbf{h}_u^{(k)} = f^{(k)} \left( \mathbf{h}_u^{(k-1)}, agg^{(k)}(\{\mathbf{h}_v^{(k-1)}, \forall v \in \mathcal{N}(u)\}) \right), \tag{1}$$

where $f$ and $agg$ are neural networks, and $\mathcal{N}(u)$ is the set of $u$'s neighbouring nodes. The superscripts are used for distinguishing the embeddings and functions at different iterations. During each message-passing iteration, a hidden representation $\mathbf{h}_u^{(k)}$ for each node $u \in V$ is updated according to the message aggregated from $v$'s neighbouring nodes. The embeddings at $k = 0$ are initialized to the node-level features, i.e., $\mathbf{h}_u^{(0)} = \mathbf{x}_u, \forall u \in V$. After $k$ iterations of message passing, every node embedding contains information about its $k$-hop neighborhood.

### 3.2 The Over-smoothing Problem

While message passing is at the heart of current graph convolutional networks, this paradigm also has major bottlenecks. Studies continually show that over-smoothing is a common issue with current message-passing graph convolutional networks. The intuitive idea of over-smoothing is that after repeatedly applying message passing, the representations for all nodes in the graph can become very similar to one another, therefore node-specific features become lost. Due to this issue, it is impossible to build deeper models to capture the longer-term dependencies in the graph.

From the perspective of graph signal processing, the graph convolution of the GCN model (Kipf & Welling, 2016) can be seen as a special form of Laplacian smoothing (Li et al., 2018) that basically updates the embedding for a node using the weighted average of the node's itself and its neighbour embeddings. But after applying too many rounds of Laplacian smoothing, the representations for all nodes will become indistinguishable from each other.

Formally, the issue of over-smoothing can be described through defining the influence of each node's input feature on the final layer embedding of all the other nodes in the graph. For any pair of node $u$ and node $v$, the influence of node $u$ on node $v$ in a graph convolutional network model can be quantified by examining the magnitude of the corresponding Jacobian matrix (Xu et al., 2018) as follows:

$$I_K(u, v) = \mathbf{1}^\top \left( \frac{\partial \mathbf{h}_v^{(K)}}{\partial \mathbf{h}_u^{(0)}} \right) \mathbf{1}, \tag{2}$$

where $\mathbf{1}$ is a vector of ones. $I_K(u, v)$, which is the sum of the entries in the Jacobian matrix $\frac{\partial \mathbf{h}_v^{(K)}}{\partial \mathbf{h}_u^{(0)}}$, is a measure of how much the initial embedding of node $u$ influences the final embedding of node $v$. Given the above definition of influence, Xu et al. (Xu et al., 2018) prove the following theorem:

**Theorem 1**. For any graph convolutional network model which uses a self-loop update approach and an aggregation function of the following form:

$$agg(\{\mathbf{h}_v, \forall v \in \mathcal{N}(u) \cup \{u\}\}) = \frac{1}{g_n(|\mathcal{N}(u) \cup \{u\}|)} \sum_{v \in \mathcal{N}(u) \cup \{u\}} \mathbf{h}_v, \tag{3}$$

where $g_n$ a normalization function, we have the following:

$$I_K(u, v) \propto p_{G,K}(u|v), \tag{4}$$

where $p_{G,K}(u|v)$ denotes the probability of visiting node $v$ on a length of $K$ random walk starting from node $u$.

Theorem 1 states that when we are using a $K$-layer graph convolutional network model, the influence of node $u$ on node $v$ is proportional to the probability of reaching node $v$ on a $K$-step random walk starting from node $u$. The consequence of this is that as $K \to \infty$ the influence of every node approaches the stationary distribution of random walks over the graph, therefore the information from local neighborhood is lost. Theorem 1 applies directly to the models that use a self-loop update approach, but the result can also be generalized asymptotically for the models that use the basic message passing update in Equation 1.

### 3.3 Proposed Method: RandAlign Regularization

Over-smoothing is a common issue in message-passing graph convolutional networks. This issue occurs when the generated node embeddings become over-smoothed and therefore uninformative after repeatedly applying message passing iterations. This is problematic because information from local neighbourhood becomes lost when more layers of message passing are added to the model. Due to the over-smoothing issue, it is difficult to stack deeper graph convolutional layers to capture long-term dependencies of the graph.

When the problem of over-smoothing occurs, the embeddings of all nodes become very similar to one another (Hamilton, 2020). Studies (Cai & Wang, 2020; Oono & Suzuki, 2019) show that successive iteration of message passing lead to node embeddings converging to the same eigenspace, which implies that the cosine similarity between one another may approach each other. The SSFG method (Zhang et al., 2022) stochastically scales the norms of the learned embeddings at each layer to improve the generalization performance of graph neural networks. This method does not explicitly address the issue of the cosine similarities converging to the similar value.

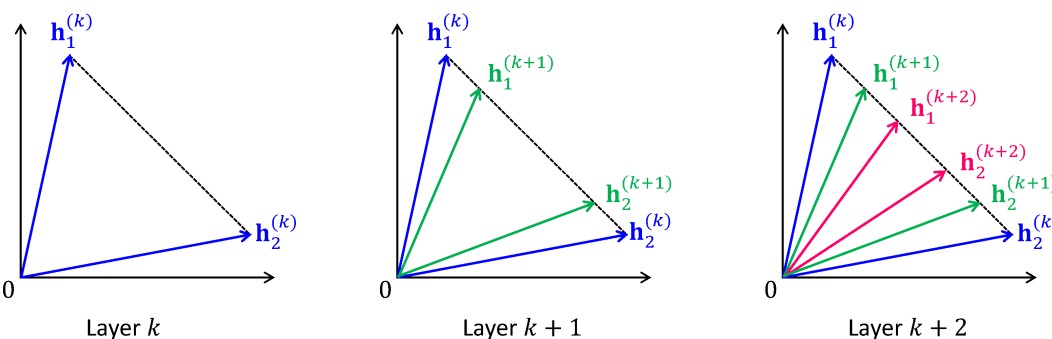

Figure 1: An illustrative example for understanding the over-smoothing issue. We consider a two node fully connected graph and use a GAT model that layerwisely learn embeddings using the equation $\mathbf{h}_i^{(k)} = \sum_{v \in \mathcal{N}(u)} \alpha_{u,v} \mathbf{h}_v^{(k-1)}$, wherein $\alpha_{u,v} > 0$ and $\sum_{v \in \mathcal{N}(u)} \alpha_{u,v} = 1$. We have simplified the model by removing the non-linearity and learnable parameter matrix. We show that the learned embeddings layerwisely become smoothed than the previous layer due to the convex combination of neighbourhood features.

As discussed above, the issue of over-smoothing occurs after applying too many layers of message passing. Intuitively, we can expect the learned embeddings for all nodes in the graph become smoothed layerwise or asymptotically layerwise. Based on this intuition, we propose RandAlign which randomly aligns the learned embedding for each node with that generated by the previous layer for regularizing graph convolutional networks. Here, we first show an example to demonstrate the intuition that each layer of graph convolution produces a smoothed version of the input. Consider we have a two node fully connected graph and use the GAT model (Veličković et al., 2018) to generate node embeddings (see Figure 1). The GAT model updates the embedding for node $u$ at the $k$-th layer of message passing using a weighted sum of information from its neighbours as follows:

$$\mathbf{h}_u^{(k)} = \sum_{v \in \mathcal{N}(u)} \alpha_{u,v}^{(k)} \mathbf{W}^{(k)} \mathbf{h}_v^{(k-1)}, \tag{5}$$

where $\alpha_{u,v}^{(k)}$ is the attention weight on neighbour $v \in \mathcal{N}(u)$ when aggregating information at node $u$, and $\mathbf{W}^{(k)}$ is a learnable weight matrix. Note that we have simplified the model by removing the non-linearity as compared to the original GAT (Veličković et al., 2018). The attention weight $\alpha_{u,v}^{(k)}$ is defined using the softmax function as follows:

$$\alpha_{u,v}^{(k)} = \frac{\exp\left(\mathbf{a}^{(k)^\top}\left[\mathbf{W}^{(k)}\mathbf{h}_u^{(k-1)} \,\|\, \mathbf{W}^{(k)}\mathbf{h}_v^{(k-1)}\right]\right)}{\sum_{v' \in \mathcal{N}(u)} \exp\left(\mathbf{a}^{(k)^\top}\left[\mathbf{W}^{(k)}\mathbf{h}_u^{(k-1)} \,\|\, \mathbf{W}^{(k)}\mathbf{h}_{v'}^{(k-1)}\right]\right)}, \tag{6}$$

where $\mathbf{a}^{(k)}$ is learnable vector, and $\|$ denotes the concatenation operator. With the softmax function, the attention weights are normalized to 1, i.e., $\sum_v \alpha_{u,v} = 1$. Therefore, the learned embedding $\mathbf{h}_u^{(k)}$ is essentially a convex combination of the information from $u$'s neighbours. As shown in Figure 1, $\mathbf{h}_1^{(k+1)}$ and $\mathbf{h}_2^{(k+1)}$ are on the dash line between $\mathbf{h}_1^{(k)}$ and $\mathbf{h}_2^{(k)}$, and $\mathbf{h}_1^{(k+2)}$ and $\mathbf{h}_2^{(k+2)}$ are on the dash line between $\mathbf{h}_1^{(k+1)}$ and $\mathbf{h}_2^{(k+1)}$. Thus, each layer of the message passing makes the generated embeddings more smoothed than those generated by the previous layer. As more message passing iterations are applied, the learned embeddings become indistinguishable from each other and thus the information from local neighbours become lost.

When the embeddings become smoothed, the average cosine similarity between one and another is reduced compared to that of the embedddings generated by the previous layer. As shown in Figure 1, the cosine between $\mathbf{h}_1^{(k+1)}$ and $\mathbf{h}_2^{(k+1)}$ is small as compared to the cosine between $\mathbf{h}_1^{(k)}$ and $\mathbf{h}_2^{(k)}$, and the cosine between $\mathbf{h}_1^{(k+2)}$ and $\mathbf{h}_2^{(k+2)}$ is small as compared to the cosine between $\mathbf{h}_1^{(k+1)}$ and $\mathbf{h}_2^{(k+1)}$. To reduce the smoothness of the generated embeddings, we randomly align the generated embedding for each node with that generated by the previous layer. Specifically, in each layer we first apply the message passing in Equation (1) to generate an intermediate embedding for each node $u \in V$:

$$\overline{\mathbf{h}}_u^{(k)} = \text{BN}\left(f^{(k)}\left(\mathbf{h}_u^{(k-1)}, agg^{(k)}(\{\mathbf{h}_v^{(k-1)}, \forall v \in \mathcal{N}(u)\})\right)\right), \tag{7}$$

where BN denotes a batch normalization layer. Then we align $\overline{\mathbf{h}}_u^{(k)}$ with $\mathbf{h}_u^{(k-1)}$ using random interpolation. To better maintain the benefit yielded by message passing in the aligned embedding, we first rescale $\mathbf{h}_u^{(k-1)}$ to have the same norm as $\overline{\mathbf{h}}_u^{(k)}$, then we apply a random interpolation between the two embeddings. Finally, the embedding for node $u \in V$ is updated with the residual connection (He et al., 2016) as follows:

$$\begin{aligned}
\mathbf{h}_u^{(k)} &= \mathbf{h}_u^{(k-1)} + align(\mathbf{h}_u^{(k-1)}, \overline{\mathbf{h}}_u^{(k)}) \\
&= \mathbf{h}_u^{(k-1)} + \lambda \frac{\mathbf{h}_u^{(k-1)}}{\|\mathbf{h}_u^{(k-1)}\|} \|\overline{\mathbf{h}}_u^{(k)}\| + (1-\lambda)\overline{\mathbf{h}}_u^{(k)},
\end{aligned} \tag{8}$$

where $align$ is a function for aligning $\overline{\mathbf{h}}_u^{(k)}$ with $\mathbf{h}_u^{(k-1)}$, and $\lambda \sim U(0,1)$ is sampled from the standard uniform distribution. By this way, we can keep the representation ability yielded by message passing while reducing the smoothness in the aligned embeddings. Because the expected value of $\lambda$ is 0.5, i.e., $E[\lambda] = 0.5$, at test time $\lambda$ is set to a fixed value of $0.5$. Algorithm 1 shows the embedding generation algorithm with the message-passing framework and our RandAlign regularization method.

---

**Algorithm 1** The embedding generation process with the message-passing framework and our RandAlign regularization method.

---

**Input:** Graph $G = (V, E)$; number of graph convolutional layers $K$; input node features $\{\mathbf{x}_v, \forall u \in V\}$

**Output:** Node embeddings $\mathbf{h}_u^{(K)}$ for all $u \in V$

1: $\mathbf{h}_u^{(0)} \leftarrow \mathbf{x}_u, \forall u \in V$
2: **for** $k = 1, ..., K$ **do**
3:      **for** $u \in \mathcal{V}$ **do**
4:          $\overline{\mathbf{h}}_u^{(k)} = \text{BN}\left(f^{(k)}\left(\mathbf{h}_u^{(k-1)}, agg^{(k)}(\{\mathbf{h}_v^{(k-1)}, \forall v \in \mathcal{N}(u)\})\right)\right)$ // generate an intermediate embedding for $u$ using a general message passing model (see Equation (7)).
5:      **end for**
6:      **for** $v \in \mathcal{V}$ **do**
7:          **if** model.training == True **then**
8:              $\lambda \sim U(0, 1)$
9:          **else**
10:              $\lambda = 0.5$
11:          **end if**
12:          $\mathbf{h}_u^{(k)} = \mathbf{h}_u^{(k-1)} + \lambda \cdot \frac{\mathbf{h}_u^{(k-1)}}{\|\mathbf{h}_u^{(k-1)}\|} \cdot \|\overline{\mathbf{h}}_u^{(k)}\| + (1 - \lambda) \cdot \overline{\mathbf{h}}_u^{(k)}$ // update the embedding for $u$ as sum of the aligned embedding and the input node embedding (see Equation (8)).
13:      **end for**
14: **end for**

---

The proposed RandAlign method is straightforward to understand. By aligning the learned embeddings with those generated by the previous layer, the smoothness of these learned embeddings is explicitly reduced, therefore the overall model performance is improved. Because the embeddings before alignment are learned using the basic message-passing framework, our RandAlign is a general method that can be applied in different message passing graph convolutional network models to alleviate the over-smoothing problem. Moreover, our RandAlign method does not introduce additional hyper-parameters or trainable weights, it can be directly applied in a plug and play manner and without the time-consuming hyper-parameter tuning procedure.

## 4 Experiments

### 4.1 Datasets and Setup

**Datasets**. The proposed RandAlign method is evaluated on four graph domain tasks: graph classification, node classification, multi-label graph classification and binary graph classification. The experiments are conducted on seven benchmark datasets, which are briefly introduced as follows.

- **MNIST and CIFAR10** (Dwivedi et al., 2020) are two datasets for superpixel graph classification. The original images in MNIST (LeCun et al., 1998) and CIFAR10 (Krizhevsky et al., 2009) are converted to superpixel graphs using the SLIC technique (Achanta et al., 2012). Each superpixel represents a small region of homogeneous intensity in the original image.

- **PascalVOC-SP** (Dwivedi et al., 2022) is also a dataset for superpixel graph classification. There are 11,355 graphs with a total of 5.4 million nodes in PascalVOC-SP. Each superpixel graph corresponds to an image in Pascal VOC 2011. The superpixel graphs in PascalVOC-SP are much large compared to those in MNIST and CIFAR10 (Dwivedi et al., 2020).

- **PATTERN and CLUSTER** (Dwivedi et al., 2020). The two datasets are used for inductive node classification. The graphs in the two datasets are generated using the stochastic block model (Abbe, 2017). PATTERN is used for evaluating the model for recognizing specific predetermined subgraphs, and CLUSTER is used for identifying community clusters in the semi-supervised setting.

- **Peptides-func** (Dwivedi et al., 2022) is a dataset of peptides molecular graphs. The nodes in the graphs represent heavy (non-hydrogen) atoms of the peptides, and the edges represent the bonds between these

Table 1: Results for superpixel graph classification on MNIST and CIFAR10. We show that our RandAlign consistently improves the performance of the base graph convolutional network models. Residual connection and batch normalization, which are simple strategies that can help to alleviate over-smoothing, are applied to the GCN and GAT base models.

| Model | MNIST | | | | |
|-------|------|---------|---------|----------|----------|
| | Mode | 4 layers | 8 layers | 12 layers | 16 layers |
| GCN | Training | 97.196±0.223 | 99.211±0.421 | 99.862±0.043 | 99.697±0.029 |
| GCN + RandAlign | | 88.311±0.262 | 92.450±0.170 | 94.283±0.192 | 95.505±0.154 |
| GCN | Test | 90.705±0.218 | 90.847±0.078 | 91.263±0.216 | 91.147±0.185 |
| **GCN + RandAlign** | | **90.305±0.140** | **92.688±0.046** | **93.470±0.035** | **94.051±0.052** |
| GAT | Training | 99.994±0.008 | 100.00±0.000 | 100.00±0.000 | 100.00±0.000 |
| GAT + RandAlign | | 96.853±0.236 | 98.492±0.294 | 99.146±0.104 | 99.189±0.158 |
| GAT | Test | 95.535±0.205 | 96.065±0.093 | 96.288±0.049 | 96.526±0.041 |
| **GAT + RandAlign** | | **96.513±0.075** | **97.250±0.049** | **97.505±0.029** | **97.553±0.034** |
| GatedGCN | Training | 100.00±0.000 | 100.00±0.000 | 100.00±0.000 | 100.00±0.000 |
| GatedGCN + RandAlign | | 99.713±0.094 | 99.933±0.048 | 99.849±0.020 | 99.813±0.023 |
| GatedGCN | Test | 97.340±0.143 | 97.950±0.023 | 98.108±0.021 | 98.132±0.022 |
| **GatedGCN + RandAlign** | | **98.120±0.076** | **98.463±0.079** | **98.494±0.054** | **98.552±0.023** |

| Model | CIFAR10 | | | | |
|-------|------|---------|---------|----------|----------|
| | Mode | 4 layers | 8 layers | 12 layers | 16 layers |
| GCN | Training | 69.523±1.948 | 77.546±0.813 | 81.073±1.224 | 84.279±0.656 |
| GCN + RandAlign | | 59.798±0.324 | 65.405±0.603 | 66.711±0.338 | 70.919±0.522 |
| GCN | Test | **55.710±0.381** | 54.242±0.454 | 53.867±0.090 | 53.353±0.184 |
| **GCN + RandAlign** | | 55.275±0.165 | **57.145±0.202** | **57.603±0.157** | **57.736±0.162** |
| GAT | Training | 89.114±0.499 | 99.561±0.064 | 99.972±0.005 | 99.980±0.003 |
| GAT + RandAlign | | 74.522±1.179 | 81.071±0.596 | 81.511±0.464 | 79.962±0.142l |
| GAT | Test | 64.223±0.455 | 64.452±0.303 | 64.423±0.121 | 64.340±0.146 |
| **GAT + RandAlign** | | **65.385±0.074** | **69.158±0.438** | **69.707±0.350** | **69.920±0.082** |
| GatedGCN | Training | 94.553±1.018 | 99.983±0.006 | 99.995±0.003 | 99.995±0.004 |
| GatedGCN + RandAlign | | 77.784±0.799 | 83.552±0.570 | 86.779±0.520 | 90.903±0.785 |
| GatedGCN | Test | 67.312±0.311 | 69.808±0.421 | 68.417±0.262 | 70.007±0.165 |
| **GatedGCN + RandAlign** | | **72.075±0.154** | **75.015±0.177** | **76.135±0.248** | **76.395±0.186** |

atoms. The graphs are categorized into 10 classes based on the peptide functions, e.g., antibacterial, antiviral, cell-cell communication. This dataset is used for evaluating the model for multi-label graph classification.

- **OGBG-molhiv** is a molecule graph dataset introduced in the open graph benchmark (OGB) (Hu et al., 2020). The nodes and edges in the graphs represent atoms and the chemical bonds between these atoms. This dataset is used for evaluating the model's ability to predict if or not the molecule can inhibit HIV virus replication, which is a binary class classification task.

**Implementation Details**. We closely follow the experimental setup as Dwivedi et al. (Dwivedi et al., 2020) and Rampasek et al. (Rampasek et al., 2022) for training the models. We use the same train/validation/test split of each dataset and report the mean and standard deviation over 10 runs. For experiments on MNIST, CIFAR10, PATTERN and CLUSTER, the Adam algorithm (Kingma & Ba, 2014) is used for optimizing the models. The learning rate is initialized to $10^{-3}$ and reduced by a factor of 2 if the loss has not improved for a number of epochs (10, 20 or 30). The training procedure is terminated when the learning rate is reduced to smaller than $10^{-6}$. For experiments on PascalVOC-SP, Peptides-func and OGBG-molhiv, the AdamW algorithm (Loshchilov & Hutter, 2017) with cosine learning rate schedule is used for training the models. The training epochs are set to 300, 200 and 150, respectively.

Table 2: Comparison with previous methods on MNIST and CIFAR10 on superpixel graph classification.

| Model | MNIST | CIFAR10 |
|---|---|---|
| GCN (Kipf & Welling, 2016) | 90.705±0.218 | 55.710±0.381 |
| MoNet (Monti et al., 2017) | 90.805±0.032 | 54.655±0.518 |
| GraphSAGE (Hamilton et al., 2017) | 97.312±0.097 | 65.767±0.308 |
| GIN (Xu et al., 2019) | 96.485±0.252 | 55.255±1.527 |
| GCNII (Chen et al., 2020) | 90.667±0.143 | 56.081±0.198 |
| PNA (Corso et al., 2020) | 97.94±0.12 | 70.35±0.63 |
| DGN (Beaini et al., 2021) | – | 72.838±0.417 |
| CRaWl (Toenshoff et al., 2021) | 97.944±0.050 | 69.013±0.259 |
| GIN-AK+ (Zhao et al., 2021) | – | 72.19±0.13 |
| 3WLGNN (Maron et al., 2019) | 95.075±0.961 | 59.175±1.593 |
| EGT (Hussain et al., 2022) | 98.173±0.087 | 68.702±0.409 |
| GPS (Bresson & Laurent, 2017) | 98.051±0.126 | 72.298±0.356 |
| GatedGCN + SSFG (Zhang et al., 2022) | 97.985±0.032 | 71.938±0.190 |
| EdgeGCN (Zhang et al., 2023) | 98.432±0.059 | 76.127±0.402 |
| Exphormer (Shirzad et al., 2023) | 98.550±0.039 | 74.754±0.194 |
| TIGT (Choi et al., 2024) | 98.230±0.133 | 73.955±0.360 |
| GAT-16 (Veličković et al., 2018) | 95.535±0.205 | 64.223±0.455 |
| **GAT-16 + RandAlign** | **97.553±0.034** (2.11%↑) | **69.920±0.082** (8.87%↑) |
| GatedGCN-16 (Bresson & Laurent, 2017) | 97.340±0.143 | 67.312±0.311 |
| **GatedGCN-16 + RandAlign** | **98.512±0.033** (1.20%↑) | **76.395±0.186** (13.49%↑) |

More details about the seven datasets, including the dataset sizes and splits, and the evaluation metrics are in the appendix section.

## 4.2 Experimental Results

**Superpixel Graph Classification on MNIST and CIFAR10**. The quantitative results on MNIST and CIFAR10 for superpixel graph classification are reported in Table 1. We experiment with three different base models: GCN, GAT and GatedGCN. We also applied residual connections (He et al., 2016) and batch normalizations (Ioffe & Szegedy, 2015) to the base models of GCN and GAT. Residual connection and batch normalization are simple strategies which are empirically helpful to reduce the over-smoothing issue and improve the numerical stability in optimization. GatedGCN employs the gated update approach in aggregating information from neighbours and also integrates residual connections and batch normalizations. We see that the base models only slightly improve the performance or see a reduced performance as the number of layers increases from 4 to 16. Without residual connection and batch normalization, the performance would drop considerably with increased layers due to over-smoothing. By integrating the RandAlign regularization method into the models, the performance of the base models consistently improves as the number of layers increases. RandAlign on GatedGCN with 16 layers yields a 6.388% performance improvement on CIFAR10, which is a 9.13% relative improvement. We also see that for the 4 layer GCN model, applying RandAlign could not improve the performance on the two datasets. This is because the model does not suffer the over-smoothing issue at this layer.

We see from Table 1 that the base models suffer serious over-fitting problem on the two datasets. For example, the GAT and GatedGCN with 8 or more graph convolutional layers obtain nearly 100% training accuracy on CIFAR10, but their test accuracy is below 70.007%. By using our RandAlign regularization method, we see that all their training accuracy reduces while the task performance improves. This shows that through tackling the over-smoothing issue with our RandAlign, the over-fitting problem is significantly reduced, and therefore the model generalization performance is improved. Figure 2 demonstrates the learning curves of the three base models with 16 layers on the CIFAR10.

Table 2 compares the performance of our results with the recent methods on MNIST and CIFAR10. EGT Hussain et al. (2022), which integrates an additional edge channels into the Transformer model and also uses global self-attention to generate embeddings, achieves 98.173% accuracy on MNIST, which is the best among the previous models. Our

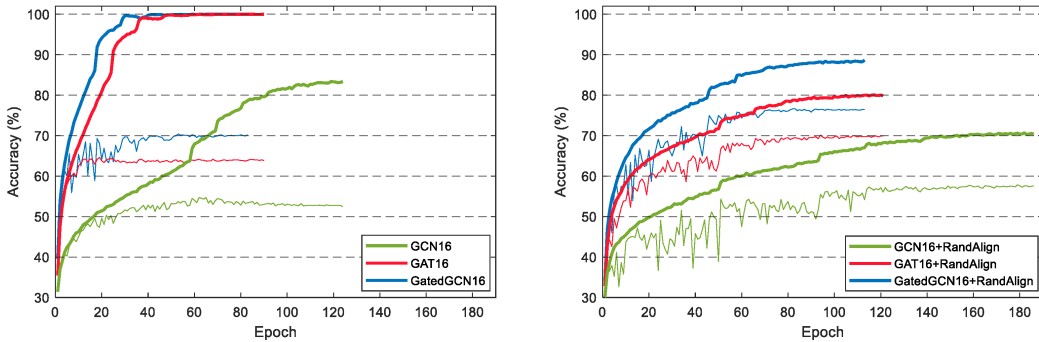

Figure 2: Learning curves on CIFAR10. Bold lines are training curves and thin lines are test curves. We show that our RandAlign method improves the generalization performance by reducing the issue of over-smoothing.

Table 3: Comparison with previous work on PascalVOC-SP on the superpixel graph classification task.

| Model | PascalVOC-SP | COCO-SP |
|---|---|---|
| GCN (Kipf & Welling, 2016) | 0.1268±0.0060 | 0.0841±0.0010 |
| GINE (Hu et al., 2019) | 0.1265±0.0076 | 0.1339±0.0044 |
| GatedGCN (Bresson & Laurent, 2017) | 0.2873±0.0219 | 0.2641±0.0045 |
| GatedGCN + RWSE (Dwivedi et al., 2021) | 0.2860±0.0085 | 0.2574±0.0034 |
| Transformer + LapPE (Dwivedi et al., 2022) | 0.2694±0.0098 | 0.2618±0.0031 |
| SAN + LapPE (Dwivedi et al., 2022) | 0.3230±0.0039 | 0.2592±0.0158 |
| SAN + RWSE (Dwivedi et al., 2022) | 0.3216±0.0027 | 0.2434±0.0156 |
| DREW (Gutteridge et al., 2023) | 0.3314±0.0024 | – |
| Exphormer (Shirzad et al., 2023) | 0.3960±0.0027 | 0.3430 ±0.0008 |
| CRAWL (Tönshoff et al., 2023) | 0.4588±0.0079 | – |
| GPS (Rampasek et al., 2022) | 0.3748±0.0109 | – |
| **GPS + RandAlign** | **0.4188±0.0062** (11.7%↑) | – |
| Fine-tuned GPS (Tönshoff et al., 2023) | 0.4440±0.0065 | 0.3884±0.0055 |
| **Fine-tuned GPS + RandAlign** | **0.4653±0.0791** (4.80%↑) | **0.3956±0.0018** (1.85%↑) |

model achieves 0.337% improved performance compared with EGT (Hussain et al., 2022). On CIFAR10, our model outperforms the previous best model DGN (Beaini et al., 2021) by 3.557%, which is a significant improvement. Our RandAlign also outperforms the SSFG regularization method (Zhang et al., 2022), which essentially stochastically scales features and gradients during training for regularization graph neural network models, but the SSFG method involves a time-consuming parameter tuning process. To the best of our knowledge, our method achieves the state of the art results on the two datasets.

**Results on PascalVOC-SP**. PascalVOC-SP is a long range superpixel classification dataset as compared to MNIST and CIFAR10. Table 3 reports the results on this dataset. We experiment with GPS (Rampasek et al., 2022) as the base model. The GPS model uses a graph convolutional network and a Transformer to model local and global dependencies in the graph. This model archives the best performance among the baseline models. We see from Table 3 that our RandAlign improves the performance of GPS from 37.48% to 42.88%, which is a 14.41% relative improvement. Once again, our RandAlign method improves the performance of the base model, advancing the state of the art result for long range graph representation learning on this dataset.

**Node Classification on PATTERN and CLUSTER**. Table 4 reports the experimental results on PATTERN and CLUSTER on node classification. We experiment with two state of the art base architectures: the spectral attention network (SAN) (Kreuzer et al., 2021) and GPS (Rampasek et al., 2022). SAN utilizes an invariant aggregation of Laplacian's eigenvectors for position encoding and also utilizes conditional attention for the real and virtual edges to improve the performance. As introduced above, a GPS layer integrates a message passing graph convolutional layer and a Transformer layer to learn local and global dependencies. We see from Table 4 our RandAlign regularization

Table 4: Experimental results PATTERN and CLUSTER on the node classification task.

| Model | PATTERN | CLUSTER |
|---|---|---|
| GCN (Kipf & Welling, 2016) | 71.892±0.334 | 68.498±0.976 |
| GraphSAGE (Hamilton et al., 2017) | 50.492±0.001 | 63.844±0.110 |
| GIN (Xu et al., 2019) | 85.387±0.136 | 64.716±1.553 |
| GAT (Veličković et al., 2018) | 78.271±0.186 | 70.587±0.447 |
| RingGNN (Chen et al., 2019) | 86.245±0.013 | 42.418±20.063 |
| MoNet (Monti et al., 2017) | 85.582±0.038 | 66.407±0.540 |
| GatedGCN (Bresson & Laurent, 2017) | 85.568±0.088 | 73.840±0.326 |
| DGN (Beaini et al., 2021) | 86.680 ± 0.034 | – |
| K-Subgraph SAT (Chen et al., 2022a) | 86.848±0.037 | 77.856±0.104 |
| GatedGCN + SSFG (Zhang et al., 2022) | 85.723±0.069 | 75.960±0.020 |
| SAN (Kreuzer et al., 2021) | 86.581±0.037 | 76.691±0.650 |
| **SAN + RandAlign** | **86.770±0.067** | **77.847±0.073** |
| GPS (Rampasek et al., 2022) | 86.685±0.059 | 78.016±0.180 |
| **GPS + RandAlign** | **86.858±0.010** | **78.592±0.052** |

Table 5: Experimental results on Peptides-func on the multi-label graph classification task.

| Model | AP (↑) |
|---|---|
| GCN (Kipf & Welling, 2016) | 0.5930±0.0023 |
| GINE (Hu et al., 2019) | 0.5498±0.0079 |
| GatedGCN (Bresson & Laurent, 2017) | 0.5864±0.0077 |
| GatedGCN + RWSE (Dwivedi et al., 2021) | 0.6069±0.0035 |
| Transformer + LapPE (Dwivedi et al., 2022) | 0.6326±0.0126 |
| SAN + LapPE (Dwivedi et al., 2022) | 0.6384±0.0121 |
| SAN + RWSE (Dwivedi et al., 2022) | 0.6439±0.0075 |
| Exphormer (Shirzad et al., 2023) | 0.6527±0.0043 |
| GPS (Rampasek et al., 2022) | 0.6535±0.0041 |
| **GPS + RandAlign** | **0.6630±0.0005** (1.45%↑) |

method improves the performance of the two base models and advances the state of the results on the two datasets. It improves the performance by 1.156% on SAN and 0.576% on GPS on the CLUSTER dataset. Our model achieves considerably improved performance when compared with GCN, GAT and GraphSAGE. Notably, the GPS model with RandAlign regularization outperforms all the baseline models on the two datasets.

**Multi-label Graph Classification on Peptides-func**. Table 5 reports the results on Peptides-func. This dataset was introduced to evaluate a model's ability to capture long-range dependencies in the graph. We also experiment with GPS (Rampasek et al., 2022) as the base model. As aforementioned, the GPS model combines a Transformer layer with the message passing graph convolutional network framework to capture the global dependencies. We see from Table 5 that our RandAlign improves the average precision of GPS from 0.6535 to 0.6630, outperforming all the baseline models including GatedGCN, Transformer (Vaswani et al., 2017) and SAN. Peptides-struct is also a long range graph dataset, as with the PascalVOC-SP dataset. The results on the two datasets also show that RandAlign helps to improve the performance for capturing long-range dependencies in the graph in graph representation learning.

**Binary Graph Classification on OGBG-molhiv**. The results on OGBG-molhiv are reported in Table 6. As with Rampasek et al. (Rampasek et al., 2022), we only compare with the baseline models that are trained from scratch. We experiment using GPS as the base model. It can be seen that RandAlign improves the ROC-AUC of GPS from 0.6535 to 0.6630, which is a relative 1.45% improvement, outperforming all the baseline models, including PNA (Corso et al., 2020), DGN (Beaini et al., 2021) and GIN-AK+ (Zhao et al., 2022).

Table 6: Experimental results on OGBG-molhiv on binary graph classification. The models are all trained from scratch.

| Model | ROC-AUC ($\uparrow$) |
|---|---|
| GCN (Kipf & Welling, 2016) | 0.7599±0.0119 |
| GIN (Xu et al., 2019) | 0.7707±0.0149 |
| PNA (Corso et al., 2020) | 0.7905±0.0132 |
| DeeperGCN (Li et al., 2020) | 0.7858±0.0117 |
| DGN (Beaini et al., 2021) | 0.7970±0.0097 |
| ExpC (Yang et al., 2022) | 0.7799±0.0082 |
| GIN-AK+ (Zhao et al., 2022) | 0.7961±0.0119 |
| SAN (Kreuzer et al., 2021) | 0.7785±0.2470 |
| GPS (Rampasek et al., 2022) | 0.7880±0.0101 |
| **GPS + RandAlign** | **0.8021±0.0305** (1.79%$\uparrow$) |

Table 7: Importance of scaling embeddings of the previous layer in alignment.

| Model | MNIST | CIFAR10 |
|---|---|---|
| GAT-8 | | |
|  w/o Lrn&Align | 96.065±0.093 | 64.452±0.303 |
|  RandAlign w/o scaling | **96.977±0.021** | **66.212±0.182** |
|  **RandAlign + scaling** | **97.250±0.049** | **69.158±0.438** |
| GatedGCN-8 | | |
|  w/o RandAlign | 97.950±0.023 | 69.808±0.421 |
|  RandAlign w/o scaling | **98.247±0.018** | **74.437±0.150** |
|  **RandAlign + scaling** | **98.463±0.079** | **75.015±0.177** |

We have shown that RandAlign is a general method for preventing the over-smoothing issue. It improves the generalization performance of different graph convolutional network models and on different domain tasks. We also see from the experimental results that applying RandAlign results in a small standard deviation for most experiments compared with the base models. This suggests that RandAlign is also effective for improving numerical stability when optimizing the graph convolutional network models.

**Importance of Scaling $\mathbf{h}_u^{(k-1)}$ in Alignment**. In our RandAlign method, we first scale $\mathbf{h}_u^{(k-1)}$ to have the norm of $\frac{\mathbf{h}_u^{(k-1)}}{\|\mathbf{h}_u^{(k-1)}\|}\|\overline{\mathbf{h}}_v^{(k)}\|$ and then apply a random interpolation between the scaled feature and $\overline{\mathbf{h}}_v^{(k)}$ (see Equation 8) for aligning $\overline{\mathbf{h}}_v^{(k)}$. To show the importance of the scaling step, we further validate our method without the scaling step. The experiments are carried out on MNIST and CIFAR10 using GAT-8 and GatedGCN-8 as the base models, and Table 12 reports the comparison results. We see that applying scaling improves the performance of the two base models on the two datasets. By scaling $\mathbf{h}_u^{(k-1)}$ to have the norm of $\overline{\mathbf{h}}_v^{(k)}$, more information about $\overline{\mathbf{h}}_v^{(k)}$ is contained in the aligned representation, and therefore the task performance is improved.

**Results of without Using Batch Normalizations and Residual Connections.** Table 8 reports the results of without using batch normalizations and residual connections on MNIST and CIFAR10. We can see from Table 8 that the base models suffer from the over-smoothing problem without using batch normalizations and residual connections. By applying our RandAlign method, the phenomenon of over-smoothing is effectively reduced.

We further validated the time cost for an epoch to take a forward run on CIFAR10 and MNIST using an RTX 6000 GPU. The GatedGCN with 16 layers was used as the base model. The results are shown in Table 9. It can be seen from Table 9 that the time cost for a training epoch in forward run is only slightly increased, while the time cost for a test epoch in forward run in nearly unchanged.

Table 8: Results of our RandAlign method on the base models, wherein batch normalizations and residual connections are not applied.

| Model | MNIST | | | | |
|---|---|---|---|---|---|
| | Mode | 4 layers | 8 layers | 12 layers | 16 layers |
| GCN | Training | 90.480±0.436 | 91.607±0.257 | 93.187±0.429 | 11.235±0.000 |
| GCN + RandAlign | | 92.857±0.255 | 93.940±0.298 | 94.109±0.417 | 90.952±0.548 |
| GCN | Test | 87.590±0.336 | 86.212±0.589 | 86.135±0.208 | 11.350±0.000 |
| **GCN + RandAlign** | | **91.442±0.251** | **92.054±0.151** | **92.197±0.284** | **90.533±0.195** |
| GAT | Training | 100.00±0.000 | 100.00±0.000 | 100.00±0.000 | 100.00±0.000 |
| GAT + RandAlign | | 97.584±0.155 | 98.990±0.192 | 98.898±0.104 | 98.850±0.139 |
| GAT | Test | 94.951±0.049 | 95.558±0.059 | 95.212±0.041 | 92.813±0.257 |
| **GAT + RandAlign** | | **96.700±0.032** | **96.863±0.054** | **96.880±0.115** | **96.938±0.026** |

| Model | CIFAR10 | | | | |
|---|---|---|---|---|---|
| | Mode | 4 layers | 8 layers | 12 layers | 16 layers |
| GCN | Training | 57.461±1.223 | 57.743±1.390 | 58.590±1.013 | 10.000±0.000 |
| GCN + RandAlign | | 51.259±0.354 | 51.933±0.368 | 51.413±0.642 | 45.939±0.609 |
| GCN | Test | **48.810±1.045** | 46.686±0.314 | 45.045±0.418 | 10.000±0.000 |
| **GCN + RandAlign** | | 48.565±0.243 | **48.643±0.216** | **47.773±0.445** | **43.037±0.637** |
| GAT | Training | 91.854±1.347 | 100.00±0.000 | 100.00±0.000 | 10.000±0.000 |
| GAT + RandAlign | | 77.601±0.110 | 79.055±0.169 | 80.736±0.921 | 81.436±0.653 |
| GAT | Test | 59.636±0.169 | 56.978±0.314 | 56.712±0.011 | 10.000±0.000 |
| **GAT + RandAlign** | | **61.866±0.095** | **61.293±0.112** | **59.640±0.300** | **60.547±0.105** |

Table 9: Time cost for an epoch to take a forward run using an RTX 6000 GPU. The results are reported in seconds.

| Model | MNIST | | CIFAR10 | |
|---|---|---|---|---|
| | Training | Test | Training | Test |
| GatedGCN-16 w/o RandAlign | 79.840 | 18.280 | 89.804 | 17.654 |
| GatedGCN-16 + RandAlign | 80.387 | 18.317 | 90.930 | 17.656 |

Figure 3 shows the norms of embeddings and average cosine similarity between adjacent nodes at different GCN layers on the MNIST dataset. We see the standard deviation of the embedding norms generated by the basic GCN16 is close to 0 and the average cosine similarity is close to 1.0, which suggests that the model suffers the over-smoothing problem. By applying our RandAlign method, the phenomenon of embedding over-smoothness is effectively prevented.

We further experimented using GAT-8 on MNIST and CIFAR10 to show the importance of the residual connection and norm scaling in Eq. (8). The results are shown in Table 10. We see that the model performance drops on the two datasets, with a significant drop on the CIFAR10 dataset. The joint use of the residual connection, random interpolation and norm scaling yields the best model performance.

Figure 4 shows the results of our RandAlign method on the basic GCN model with different message-passing layers. Following up on the results in Table 8, we further experimented using 13, 14, and 15 layers. We see that our RandAlign method helps improve the generalization performance of the base model. It prevents the model performance from dropping significant with increased layers on the two datasets.

## 5 Conclusions

Over-smoothing is a common issue in message-passing graph convolutional networks. In this paper, we proposed RandAlign for regularizing graph convolutional networks through reducing the over-smoothing problem. The basic

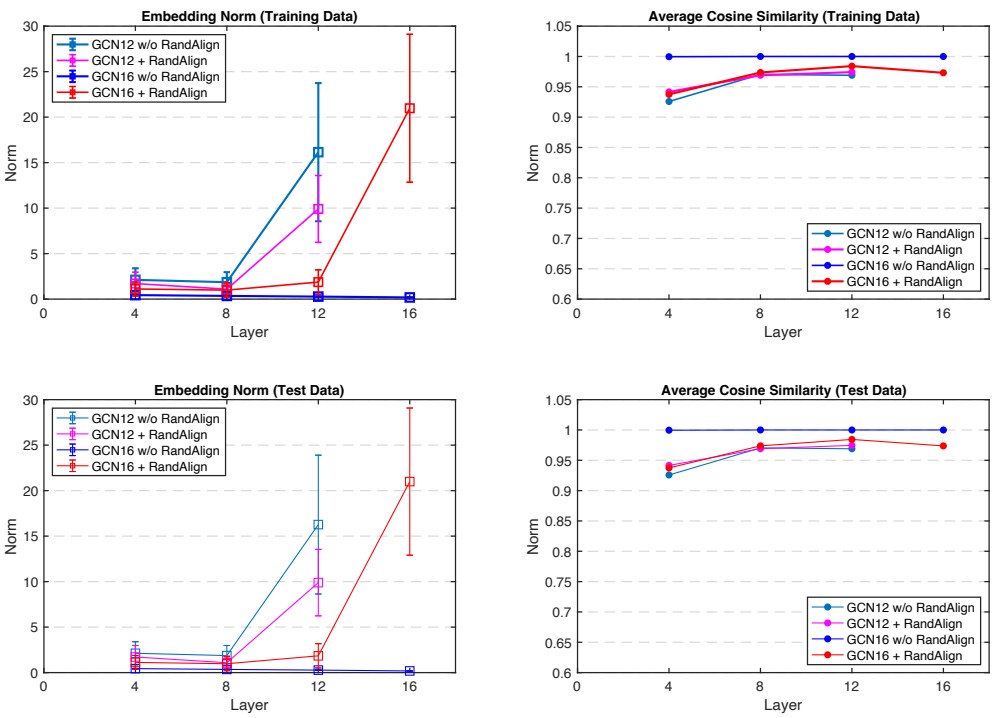

Figure 3: Norms of embeddings and average cosine similarity between adjacent nodes at different model layers on the MNIST dataset. We show the mean and and standard deviation of the embeddings generated at different model layers.

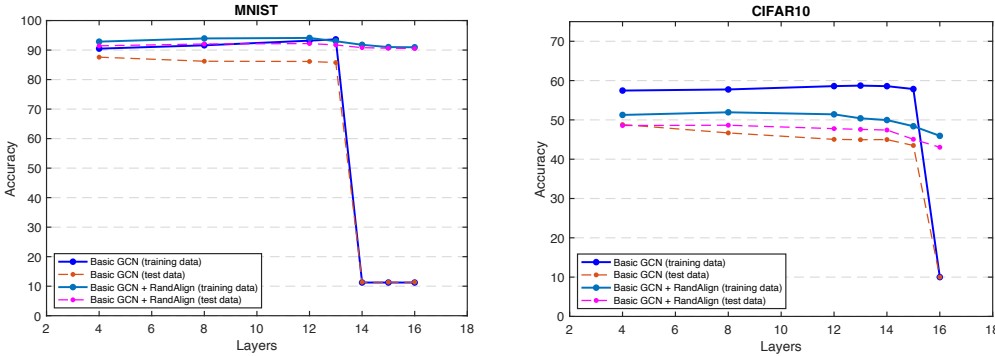

Figure 4: Performance of our RandAlign method on the basic GCN model with different message-passing layers.

idea of RandAlign is to randomly align the generate embedding for each node and with that generated by the previous layer in each message passing iteration. Our method is motivated by the intuition that learned embeddings for the nodes become smoothed layerwisely or asymptotically layerwisely. In our RandAlign, a random interpolation method is utilized for feature alignment. By aligning the generated embedding for each node with that generated by the previous layer, the smoothness of these embeddings is reduced. Moreover, we introduced a scaling step to scale the embedding of the previous layer to the same norm as the generated embedding before performing random interpolation. This scaling step can better maintain the benefit yielded by graph convolution in the aligned embeddings. The proposed RandAlign is a parameter-free method, and it can be directly applied current graph convolutional networks without introducing additional trainable weights and the hyper-parameter tuning procedure. We experimentally evaluated RandAlign on seven popular benchmark datasets on four graph domain tasks including graph classification, node classification, multi-label graph classification and binary graph classification. We presented extensive results to demonstrate RandAlign is a generic method that improves the performance of a variety of graph convolutional network models and advances the state of the art results for graph representation learning.

Table 10: Importance of residual connection and norm scaling in Eq. (8).

| Model | Residual connection | Interpolation | Norm scaling | MNIST | CIFAR10 |
|-------|---------------------|---------------|--------------|-------|---------|
| GAT-8 | − | ✓ | − | 96.786±0.188 | 62.040±0.412 |
|       | ✓ | ✓ | ✓ | 97.250±0.049 | 69.158±0.438 |

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

## A  Appendix

The details of the benchmark datasets used in the experiments are show in below Table 11.

Table 11: Details of the seven benchmark datasets used in the experiments.

| Dataset | Graphs | Nodes | Avg. nodes/graph | #Training | #Validation | #Test | #Categories | Task |
|---------|--------|-------|------------------|-----------|-------------|-------|-------------|------|
| MNIST | 70K | – | 40-75 | 55,000 | 5000 | 10,000 | 10 | Superpixel graph classification |
| CIFAR10 | 60K | – | 85-150 | 45,000 | 5000 | 10,000 | 10 | |
| PascalVOC-SP | 11,355 | 5,443,545 | 479.40 | 8,489 | 1,428 | 1,429 | 20 | Node classification |
| COCO-SP | 123,286 | 58,793,216 | 476.88 | 113,286 | 5,000 | 5,000 | 81 | |
| PATTERN | 14K | – | 44-188 | 10,000 | 2000 | 2000 | 2 | Node classification |
| CLASTER | 12K | – | 41-190 | 10,000 | 1000 | 1000 | 6 | |
| Peptides-Func | 15,535 | 2,344,859 | 150.90 | 70% | 15% | 15% | 10 | Multi-label graph classification |
| OGBG-Molhiv | 41,127 | – | 25.50 | 80% | 10% | 10% | 2 | Binary graph classification |

**Evaluation Metrics.** Following Dwivedi et al. (2020) and Rampasek et al. (2022), the following metrics are used for different domain tasks. For node classification on PATTERN and CLUSTER, the performance is measured using the weighted accuracy. Te performance on PascalVOC-SP and COCO-SP is evaluated using the macro weighted F1 score. For superpixel graph classification, we report the classification accuracy on test set. For multi-label graph classification on Peptides-func, the performance is measured using average precision (AP) across the categories. For the

Table 12: Results (%) on the Cora dataset.

| Model | 4 layers | 8 layers | 16 layers |
|---|---|---|---|
| GAT | 83.2 | 28.3 | 0.09 |
| GAT + RandAlign | 82.6 | 82.5 | 82.2 |

binary classification task on OGBG-molhiv, the performance is measured using the area under the receiver operating characteristic curve (ROC-AUC).

We further experimented on the Cora dataset using GAT as the base model. Our implementation is based on `https://github.com/Diego999/pyGAT`. The base model uses Dropout as a regularization method. We found that the joint use of Dropout and RandAlign results in reduced classification accuracy. Therefore, we removed the Dropout method in the base model. Additionally, we found that randomly aligning features with those generated by the first GAT layer yields better results. The results on the Cora dataset are reported in Table 12. We see that when the number of layers is increased to 16, the model results in significantly reduced performance. With our RandAlign method, the model performance remains almost unchanged with increased layers. The results suggest that our methods helps to alleviate the over-smoothing problem when deeper layers are used.

