# OpenReview forum: "RandAlign: A Parameter-Free Method for Regularizing Graph Convolutional Networks"
_TMLR — Rejected by TMLR_

### Review · Reviewer_iVRZ · 2024-05-25

**Summary Of Contributions:**

The paper addresses the prevalent issue of over-smoothing in message-passing graph convolutional networks (GCNs), where node embeddings become increasingly similar and less informative with each layer of message passing. To counter this, the authors introduce RandAlign, a stochastic regularization method that reduces embedding smoothness by randomly aligning each node’s embedding with that of the previous layer using random interpolation. This process includes a scaling step to ensure the benefits of graph convolution are retained in the aligned embeddings. RandAlign is parameter-free, requiring no additional trainable weights or hyper-parameters, and can be seamlessly integrated into existing GCN architectures. The method has been empirically validated across seven benchmark datasets and various graph domain tasks.

**Audience:**

Yes

**Broader Impact Concerns:**

No limitations or potential negative societal impacts are explicitly discussed.

**Claims And Evidence:**

No

**Requested Changes:**

Please kindly refer to the Strengths and Weaknesses.

**Strengths And Weaknesses:**

Strengths:
* The paper presents RandAlign, a novel approach that effectively mitigates the over-smoothing problem in GCNs.
* RandAlign does not add complexity to the model as it is parameter-free, making it an attractive option for enhancing existing GCNs without the need for additional parameter tuning.

Weaknesses:
* The benchmark tasks used in this paper are very saturated and performance increases are frequently a result of fine-grained hyperparameter tuning. It can be observed from Table 1-5 that many cases of the improvements are very marginal and also within the stds of baselines, let alone GCN + RandAlign achieves worse accuracy than GCN for 4 layers but is marked as bold in Table 1. A detailed baseline hyperparameter exploration study or a state-of-the-art result in a large-scale graph learning challenge would help clarify the significance of the results.
* The studied baselines are outdated, i.e., all before 2023. The space of handling over-smoothing issues for GNNs is very crowded with expansive literature published over the past years. More analysis and comparison with recent works are essential to demonstrate the paper’s exceptional.
* The authors claim many times in the manuscript that the SSFG method (Zhang et al., 2022) explicitly breaks the norms of the embeddings converging to a similar value regarding the second respect for reducing over-smoothing. Quantitative comparison on how RandAlign does not influence the norms of embeddings like SSFG should be analyzed to support this claim as this is the most important motivation for RandAlign.
* The stochastic nature of RandAlign, which relies on random interpolation for alignment, could introduce variability in the results. This randomness needs to be verified with experiments to ensure it wouldn’t lead to inconsistencies in performance across different runs or datasets.
* Even though RandAlign does not introduce additional trainable parameters, the process of random alignment and scaling at each layer could add computational overhead, especially for large-scale graphs. This could not be observed from current experiments since the used benchmarks are all relatively small.

---

> ### Author Response · Authors · 2024-06-20
> **Author Response**
>
> **The benchmark tasks used in this paper are very saturated and performance increases are frequently a result of fine-grained hyperparameter tuning. It can be observed from Table 1-5 that many cases of the improvements are very marginal and also within the stds of baselines. A detailed baseline hyperparameter exploration study or a state-of-the-art result in a large-scale graph learning challenge would help clarify the significance of the results.**
>
> *Response:* For fair comparison, we used the same model hyperparameters and optimizer hyperparameters as the base models, other than for results on PascalVOC-SP and COCO-SP we used a larger number of training epochs (300 vs 200). The results were not obtained by using a fine-grained tuning.
>
> For results in Table 2, Table 3, Table 5 and Table 6, we have included the relative improvements compared to the base models.
> The use of RandAlign on the 4 layer GCN yields a small reduced performance, this is because the model does not suffer the problem of over-smoothing. For the GCN model with 8 or more layers, applying RandAlign consistently improves the model performance.
> We have maked the result of the base 4-layer GCN model as bold. Additionally, we have experimented  without using batch normalizations and residual connections on the base GCN and GAT. The results are reported in Table 8, from which we can see that our RandAlign method is effective in reducing over-smoothing.
>
> We have further experimented on the COCO-SP dataset, which has 123,286 graphs with 58,793,216 nodes.
> We have validated our method on Fine-tuned GPS (Tönshoff et al., 2023), which uses extensive tuning, on PascalVOC-SP and COCO-SP. The use of RandAlign yields  a relative improvement of 4.80\%  and 1.85\% on PascalVOC-SP and COCO-SP respectively.
>
> **The studied baselines are outdated, i.e., all before 2023. The space of handling over-smoothing issues for GNNs is very crowded with expansive literature published over the past years. More analysis and comparison with recent works are essential to demonstrate the paper’s exceptional.**
>
> *Response:* We have compared our results against EdgeGCN (Zhang et al., 2023), Exphormer (Shirzad et al., 2023) and TIGT (Choi et al., 2024) on MNIST and CIFAR10. Our model (GatedGCN + RandAlign) outperforms the three baseline models on CIFAR10, and it achieves comparable performance compared to Exphormer, which is the best among the three models.
>
>  We have also compared our resutls against DREW (Gutteridge et al., 2023), Exphormer (Shirzad et al., 2023) and CRAWL (Tönshoff et al., 2023) on PascalVOC-SP and COCO-SP. Our method also achieves improved performance compared to the three baseline models. Our model (GPS + RandAlign) also outperforms Exphormer (Shirzad et al., 2023) on the Peptides-func dataset.
>
> **The authors claim many times that the SSFG method explicitly breaks the norms of the embeddings converging to a similar value regarding the second respect for reducing over-smoothing. Quantitative comparison on how RandAlign does not influence the norms of embeddings like SSFG should be analyzed to support this claim as this is the most important motivation for RandAlign.**
>
> *Response:*  We wanted to make a comparison between our method and the SSFG method. When the problem of over-smoothing occurs, the embeddings of all nodes become very similar to one another  (Hamilton, 2020). The SSFG method stochastically scales the norms of the learned embeddings at each layer during training to improve the generalization performance of graph neural networks. Unlike the SSFG method, our method is to  randomly align the generated
> embeddign for every node with that generated by the previous layer, to reduce the smoothness of the generated embeddings. The motivation of our method is based on the intuition that each layer of message-passing makes a smoothed
> version of the input.
>
> **This randomness needs to be verified with experiments to ensure it wouldn’t lead to inconsistencies in performance across different runs or datasets.**
>
> *Response:* The mean and standard deviation over 10 runs are reported as the experimental results. It can be seen from Table 1 to Table 6, our results have  smaller deviation values  in terms of training performance and test performance for most cases. This suggests that our method helps to improve the stability of optimization.
>
> **The process of random alignment and scaling could add computational overhead, especially for large-scale graphs. This could not be observed from current experiments since the used benchmarks are all relatively small.**
>
> *Response:* We have validated the time cost for an epoch to take a forward run on CIFAR10 and MNIST using an RTX 6000 GPU. The GatedGCN with 16 layers was used as the base model. The results are shown in Table 9. It can be seen from Table 9 that the time cost for a training epoch in forward run is only slightly increased, while  the time cost for a test epoch in forward run in nearly unchanged.

---

### Review · Reviewer_V4tS · 2024-05-26

**Summary Of Contributions:**

The oversmoothing issue in message-passing graph convolutional networks (GCNs) has been a persistent challenge. When repeatedly applying message passing iterations, the learned embeddings for all nodes tend to become overly similar, rendering them uninformative. To address this, the authors of this paper propose RandAlign, which randomly aligns the learned embedding for each node with the previous layer’s embedding using random interpolation. Experimental evaluations across various graph domain tasks demonstrate that RandAlign improves generalization performance.

**Audience:**

Yes

**Claims And Evidence:**

No

**Requested Changes:**

Please address my concerns in the weakness section, particularly regarding the explanation of how you conclude that this method alleviates oversmoothing. It's important to note that improving performance does not necessarily mean avoiding oversmoothing.

Also, I have identified some typos:
- In the abstract, "RndAlign" should be corrected to "RandAlign."
- In the introduction, "PariNorm" should be corrected to "PairNorm."

**Strengths And Weaknesses:**

## Strengths:
- The proposed method demonstrates a great ability to enhance generalized performance across various experimental evaluations.
- This paper provides an interesting intuition to demonstrate over-smoothing using the concept of convex hulls while it ignores the nonlinearity and learnable weights.

## Weaknesses:
- I'm concerned about the assertion that the similarity in norms among these embeddings is indicative of oversmoothing. As highlighted in [1] and [2], it's been proved that through successive iterations of message passing, node embeddings tend to converge to the same eigenspace rather than exactly the same values. This convergence implies that while the cosine similarity may approach each other, there's no guarantee that the norms will do the same.

- Could you please clarify the theoretical rationale behind the performance improvement gained by rescaling $h^{k-1}_u$ to match the norm of $\overline{h}^{k}_u$? This aspect is somewhat ambiguous to me. Scaling $h^{k-1}_u$ to match the norm of $\overline{h}^{k}_u$ doesn't inherently imply that the aligned representation will contain more proportion from $\overline{h}^{k}_u$. This assertion seems not true especially when the norm of $h^{k-1}_u$ is smaller than that of $\overline{h}^{k}_u$.

- Based on the experimental findings, I haven't observed any model experiencing oversmoothing. In fact, typically, increasing the number of layers leads to improved results. Therefore, it's not evident that this method effectively addresses the oversmoothing concern. Instead, it appears to enhance the model's generalization ability through adding stochastic biases.

[1] Cai, C., & Wang, Y. (2020). A note on over-smoothing for graph neural networks. arXiv preprint arXiv:2006.13318.
[2] Oono, K., & Suzuki, T. (2019). Graph neural networks exponentially lose expressive power for node classification. arXiv preprint arXiv:1905.10947.

---

> ### Author Response · Authors · 2024-06-20
> **Author Response**
>
> **I'm concerned about the assertion that the similarity in norms among these embeddings is indicative of oversmoothing. As highlighted in [1] and [2], it's been proved that through successive iterations of message passing, node embeddings tend to converge to the same eigenspace rather than exactly the same values. This convergence implies that while the cosine similarity may approach each other, there's no guarantee that the norms will do the same.**
>
> *Response:* Thank you for the comment. We have revised our wordings as follows. When over-smoothing occurs, the embedeings of all nodes become very similar to one another  (Hamilton, 2020). The studies of [1] and [2] show that successive iteration of message passing lead to node embeddings converging to the same eigenspace, which implies that the cosine similarity between one another may approach each other. The SSFG method (Zhang et al., 2022) stochastically scales the norms of the learned embeddings at each layer to improve the generalization performance of graph neural networks. This method does not explicitly address the issue of the cosine similairties  converging to the similar value.
>
> **Could you please clarify the theoretical rationale behind the performance improvement gained by rescaling $h^{k-1}_u$ to match the norm of $\overline{h}^{k}_u$? This aspect is somewhat ambiguous to me. Scaling $h^{k-1}_u$ to match the norm of $\overline{h}^{k}_u$ doesn't inherently imply that the aligned representation will contain more proportion from $\overline{h}^{k}_u$. This assertion seems not true especially when the norm of $h^{k-1}_u$ is smaller than that of $\overline{h}^{k}_u$.**
>
> *Response:* Without rescaling $\mathbf{h}_u^{(k-1)}$ to the norm of $\overline{\mathbf{h}}_u^{(k)}$, $align(\mathbf{h}_u^{(k-1)}, \overline{\mathbf{h}}_u^{(k)})$ equals to $ \lambda\mathbf{h}_u^{(k-1)} + (1-\lambda) \overline{\mathbf{h}}_u^{(k)}$. The norm of $ \lambda\mathbf{h}_u^{(k-1)} + (1-\lambda) \overline{\mathbf{h}}_u^{(k)}$ is:
> $n_1 = {\lambda^2{\color{blue}\lVert \mathbf{h}_u^{(k-1)}\rVert}^2} + (1-\lambda)^2 \lVert \overline{\mathbf{h}}_u^{(k)} \rVert^2 + 2\lambda(1-\lambda){\color{red}\lVert \mathbf{h}_u^{(k-1)}\rVert }\lVert \overline{\mathbf{h}}_u^{(k)} \rVert cos(\mathbf{h}_u^{(k-1)}, \overline{\mathbf{h}}_u^{(k)})$
>
> The norm of $\lambda  \frac{\mathbf{h}_u^{(k-1)}}{\lVert \mathbf{h}_u^{(k-1)}\rVert}  {\lVert \overline{\mathbf{h}}_u^{(k)}\rVert} + (1-\lambda)\overline{\mathbf{h}}_u^{(k)}$ is:
> $n_2 = {\lambda^2{\color{red}\lVert \overline{\mathbf{h}}_u^{(k)} \rVert}^2} + (1-\lambda)^2 \lVert \overline{\mathbf{h}}_u^{(k)} \rVert^2 + 2\lambda(1-\lambda){\color{red}\lVert \overline{\mathbf{h}}_u^{(k)}\rVert} \lVert \overline{\mathbf{h}}_u^{(k)} \rVert cos(\mathbf{h}_u^{(k-1)}, \overline{\mathbf{h}}_u^{(k)})$
>
> The difference between $n_1$ and $n_2$ are highlighted in red and blue. Compared with $n_1$, $n_2$ contains more information about $\overline{\mathbf{h}}_u^{(k)}$. In Eq. (8), $\mathbf{h}_u^{(k-1)}$ is scaled to the norm of $\overline{\mathbf{h}}_u^{(k)}$, then the interpolation is applied. Once the direction of the vector $\mathbf{h}_u^{(k-1)}$ is given, the $align$ function generates the same interpolated vector regardless of the original norm of $\mathbf{h}_u^{(k-1)}$.
>
> **Based on the experimental findings, I haven't observed any model experiencing oversmoothing. In fact, typically, increasing the number of layers leads to improved results. Therefore, it's not evident that this method effectively addresses the oversmoothing concern. Instead, it appears to enhance the model's generalization ability through adding stochastic biases.**
>
>  *Response:* For results in Table 1, the batch normalization and residual connections were both applied in the baseline models. The two methods help to reduce the problem of over-smoothing.
>
> For results in Table 1, both batch normalization and residual connections were  applied in the baseline models. These two methods were basic methods that help to reduce the problem of over-smoothing.
>  Therefore the problem of over-smoothing was not obvious for resutls in Table 1.
>
>  We have experimented without using batch normalizations and residual connections on the baseline models.
>      The results are reported in Table 8. We can see from Table 8 that the base models suffer from the over-smoothing problem without using batch normalizations or residual connections.
>      By applying our RandAlign method, the phenomenon of over-smoothing is effectively reduced.

---

> > ### Comment · Reviewer_V4tS · 2024-06-29
> > **Further Concerns**
> >
> > Thank you for the authors' detailed response. I appreciate their effort to address my concerns.
> >
> > However, I still have some questions regarding its effectiveness in preventing over-smoothing.
> >
> > Firstly, while the authors explain the benefits of rescaling, it's unclear how it directly addresses over-smoothing. Since cosine similarity relies on the direction of vectors, not their magnitude (norm), using norm as a reference of containing information might not be the most suitable approach.  Could the authors elaborate on how rescaling tackles over-smoothing specifically?
> >
> > Secondly, while the new empirical study is valuable, it doesn't definitively **isolate the impact of over-smoothing**. Performance gains could also be attributed to improved generalization. To provide stronger evidence against over-smoothing, an experiment similar to Table 1, **analyzing model performance with varying layer depths**, would be helpful. This would allow for a more direct observation of how the model handles increasing complexity, potentially revealing signs of over-smoothing. Ultimately, demonstrating a clear example of over-smoothing and how the proposed technique addresses it would be the most convincing approach.

---

> > > ### Author Response · Authors · 2024-07-04
> > > **Author Response**
> > >
> > > **Firstly, while the authors explain the benefits of rescaling, it's unclear how it directly addresses over-smoothing. Since cosine similarity relies on the direction of vectors, not their magnitude (norm), using norm as a reference of containing information might not be the most suitable approach. Could the authors elaborate on how rescaling tackles over-smoothing specifically?**
> > >
> > > *Response:* Aligning the learned embedding for each node with that generated by the previous layer using random interpolation is the core step to reduce the smoothness of the learned embeddings.
> > >     The use of norm rescaling enables the aligned features to maintain more information learned by the current layer of message passing, otherwise the aligned features would contain more information about the previous layer.
> > >     The norm rescaling step does not explicitly tackle the over-smoothing problem; however, the joint use random interpolation and norm rescaling helps reduce the smootheness of the generated embeddings while maintaining the  representational ability yielded by message passing.
> > >
> > > **Secondly, while the new empirical study is valuable, it doesn't definitively isolate the impact of over-smoothing. Performance gains could also be attributed to improved generalization. To provide stronger evidence against over-smoothing, an experiment similar to Table 1, analyzing model performance with varying layer depths, would be helpful. This would allow for a more direct observation of how the model handles increasing complexity, potentially revealing signs of over-smoothing. Ultimately, demonstrating a clear example of over-smoothing and how the proposed technique addresses it would be the most convincing approach.**
> > >
> > > *Response:* We have included Figure 4 (see Page 14) to show the results of our RandAlign method on the basic GCN model with different message-passing layers. Following up on the results in Table 8, we further experimented using 13, 14, and 15 layers. It can be seen from Figure 4 that our RandAlign method helps improve the generalization performance of the base model. It prevents the model performance from dropping significantly with increased layers on the two datasets.
> > >
> > >  We have also included a Figure 3 (see Page 14) to show the  norms of embeddings and average cosine similarity between adjacent nodes at different layers of the basic GCN12 model and the basic GCN16 model on the MNIST dataset. The standard deviation of the embedding norms generated by the basic GCN16 model is close to 0 and the average cosine similarity is close to 1.0, which indicates that the model suffers severer over-smoothing problem. By using our RandAlign method, the issue of embedding over-smoothness is effectively prevented.

---

### Review · Reviewer_ATuJ · 2024-06-06

**Summary Of Contributions:**

The paper introduces RandAlign, a regularization method for Graph Convolutional Networks, which aims to reduce oversmoothing. RandAlign uses random interpolation in every layer, in order to align the generated node embeddings with the ones created by the previous layer.

**Audience:**

Yes

**Claims And Evidence:**

No

**Requested Changes:**

Please kindly refer to the Weaknesses.

**Strengths And Weaknesses:**

Strengths:

- A new method to tackle an interesting problem of GNNs.

- The proposed method is parameter free and does not yield additional training cost.

- Experiments were conducted using a variety of baseline models.

$\\\\$

Weaknesses:

- Can the authors further explain the need of presenting the analysis of section 3.3 about oversmoothing and a simplified version of GAT? The presence of oversmoothing as depth increases is undeniable based on [1].

- In their presented method and in Eq. 8 the authors also utilize residual connections except of the normalization method they propose. Moreover they do not normalize the embeddings of the previous layer with any kind of norm as they do with the embeddings generated by the current layer. This in turn might lead $h^{(k-1)}_u$ dominate the resulting embeddings, hence making the contribution of current layer negligible. Can the authors further elaborate on this? Maybe some experiments with the embeddings of each layer and their proximity or distance from the embeddings of the previous layer might help understanding. Additional experiments (ablation study) using Eq. 8 without the residual connection would also shed more light to the actual effect of the proposed regularization.

- The intuition behind choosing different $\lambda$ for training and testing is not clear. Can the authors further elaborate on this and in general further explain what is the behavior of the model for different values of $\lambda$?

- Datasets used for experiments are not the usual ones seen in literature. Adding experiments with classical datasets like Cora, CiteSeer etc. would be beneficial. Also increasing model's depth would also help to better understand the effect of the proposed method.

- Table 1 shows that increasing depth does not reduce performance of baselines (without the RandAlign addition). It is not clear to me if these methods suffer from oversmoothing on these particular datasets and depths. This verifies the previous bullet and the need for further experimentation and alternative datasets and depths.

- There are several typos (e.g. RndAlign in the Abstract) and the text structure and presentation can be further improved.

[1] K. Oono and T. Suzuki. Graph neural networks exponentially lose expressive power for node classification. In 8th International Conference on Learning Representations, ICLR 2020.

---

> ### Author Response · Authors · 2024-06-20
> **Author Response**
>
> **Can the authors further explain the need of presenting the analysis of section 3.3 about oversmoothing and a simplified version of GAT?**
>
> *Response:* The main idea of our method is to randomly align the learned embedding for every node with that generated by the previous layer. Before introducing our method, we wanted to show the intution that learned embeddings for all nodes in the graph become smoothed layerwisely or asymptotically. Through the example of the simiplied version of the GAT in Figure 3, it can be visually seen how a GAT layer generates a smoothed version of the input. The analysis and example could help others better understand the idea of our method.
>
> **In their presented method and in Eq. (8) the authors also utilize residual connections except of the normalization method they propose. Moreover......of the proposed regularization.**
>
> *Response:* Eq. (8) shows the core step of our method, which is to randomly align the learned embedding for each node with that generated by the previous layer. Normalizations and residual connections empirically help to improve the overall performance. Actually, batch normalization and residual connections are both applied in our experiments on all the base models. For simplicity, the batch normalization was omitted in Eq. (8) as well as in Algorithm (1). In our revised version, we have included the use of batch normalization in Eq. (7) and Algorithm (1). We have also experimented without using batch normalization or residual connection. The results are reported in Table 8.
>
> **Can the authors further elaborate on this and in general further explain what is the behavior of the model for different values of $\lambda$?**
>
> *Response:* In our method, the value of $\lambda$ is sampled from the standard uniform distribution. The expectation of $\lambda$ equals to 0.5, i.e., $E(\lambda)=0.5$. This behavior is similar to that in the Dropout method, wherein the output of a neuron is randomly set to 0 during training and is scaled by $1/p$ at the test stage.
>
> The *align* function can be described as follows:
>  $align(\mathbf{h}_u^{(k-1)}, \overline{\mathbf{h}}_u^{(k)})=  \lambda  \frac{\mathbf{h}_u^{(k-1)}}{\lVert \mathbf{h}_u^{(k-1)}\rVert}  {\lVert \overline{\mathbf{h}}}_u^{(k)}\rVert + (1-\lambda) \overline{\mathbf{h}}_u^{(k)}$
>
> The output of the $align$ function is the interpolation between $ \frac{\mathbf{h}_u^{(k-1)}}{\lVert \mathbf{h}_u^{(k-1)}\rVert}  {\lVert \overline{\mathbf{h}}}_u^{(k)}\rVert$ and $\overline{\mathbf{h}}_u^{(k)}$, and the value of $\lambda$ is sampled from the standard uniform distribution, i.e., $U(0,1)$. Particularly, if the sampled value of $\lambda$ is 0, then  $align(\mathbf{h}_u^{(k-1)}, \overline{\mathbf{h}}_u^{(k)})$ equals to $\overline{\mathbf{h}}_u^{(k)}$. If the sampled value of $\lambda$ is 0, $align(\mathbf{h}_u^{(k-1)}, \overline{\mathbf{h}}_u^{(k)})$ equals to $\frac{\mathbf{h}_u^{(k-1)}}{\lVert \mathbf{h}_u^{(k-1)}\rVert}  {\lVert \overline{\mathbf{h}}}_u^{(k)}\rVert$, which has the same norm as $\overline{\mathbf{h}}_u^{(k)}$.
>
> **Datasets used for experiments are not the usual ones seen in literature. Adding experiments with classical datasets like Cora, CiteSeer etc. would be beneficial. Also increasing model's depth would also help to better understand the effect of the proposed method.**
>
> *Response:* We have experimented on more more dataset, COCO-SP, which has 123,286 graphs with 58,793,216 nodes. We have validated our method on Fine-tuned GPS (Tönshoff et al., 2023), which uses increased model layers compared to the base GPS model (Rampasek et al., 2022). The use of RandAlign can further improves the performance of the Fine-tuned GPS model. We have also experimented on Cora using the base GAT model. However, we found that using RandAlign does not affect the model performance.
>
> **Table 1 shows that increasing depth does not reduce performance of baselines (without the RandAlign addition). It is not clear to me if these methods suffer from oversmoothing on these particular datasets and depths. This verifies the previous bullet and the need for further experimentation and alternative datasets and depths.**
>
> *Response:* For results in Table 1, the batch normalization and residual connections were both applied in the baseline models. The two methods help to reduce the problem of over-smoothing. Therefore the problem of over-smoothing was not obvious.
>
> We have experimented without using batch normalizations and residual connections on the baseline models. The results are reported in Table 8. We can see from Table 8 that the base models suffer from the over-smoothing problem, and the ues our RandAlign helps to alleviate this phenomenon.
>
>  **There are several typos (e.g. RndAlign in the Abstract) and the text structure and presentation can be further improved.**
>
>  *Response:* We have carefully read through our manuscript to correct the typos and enhance its clarity and readability.

---

> > ### Comment · Reviewer_ATuJ · 2024-06-20
> > **Thank you for your response**
> >
> > I would like to thank the authors for their detailed response.
> >
> > Poor performance of the baselines presented in Table 8 is not necessarily due to oversmoothing. We observe poor training performance, which could be attributed to other factors beyond oversmoothing. Can the authors further verify the existence of the problem with either some visualization (e.g., t-SNE plots) or some other metrics (e.g., MADGap metric or Dirichlet energy)?
> >
> > The align(. , .) function resembles skip connection approaches, which raises the question of what would happen if one omits the residual connection and the appearing norms from Equation 8. Would this still yield similar results?
> >
> > I think that a figure depicting the norm values and how they evolve, along with a figure of the node embeddings distance metric, would improve the understanding of the proposed method.
> >
> > How can the authors explain that RandAlign does not improve the performance of GAT (which I suppose oversmooths) in Cora?

---

> > > ### Author Response · Authors · 2024-07-04
> > > **Author Response**
> > >
> > > **Poor performance of the baselines presented in Table 8 is not necessarily due to oversmoothing. We observe poor training performance, which could be attributed to other factors beyond oversmoothing. Can the authors further verify the existence of the problem with either some visualization (e.g., t-SNE plots) or some other metrics (e.g., MADGap metric or Dirichlet energy)?
> > > I think that a figure depicting the norm values and how they evolve, along with a figure of the node embeddings distance metric, would improve the understanding of the proposed method.**
> > >
> > > *Response:*  Thank you for the comment. We have included a Figure 3 (see Page 14) to show the  norms of embeddings and average cosine similarity between adjacent nodes at different  layers of the basic GCN12 model and the basic GCN16 model on the MNIST dataset. We can see from Figure 3 that the standard deviation of the embedding norms generated by the basic GCN16 is close to 0 and the average cosine similarity is close to 1.0, indicating that the model suffers the over-smoothing problem. By using our RandAlign method, the issue of embedding over-smoothness is effectively prevented.
> > >
> > > **The $align(. , .)$ function resembles skip connection approaches, which raises the question of what would happen if one omits the residual connection and the appearing norms from Equation 8. Would this still yield similar results?**
> > >
> > > *Response:* We have further experimented using GAT-8 on MNIST and CIFAR10 to show the importance of the residual connection and norm scaling in Eq. (8). The results are shown in Table 10 (see Page 15). We see that the model performance drops on the two datasets, with a significant drop on the CIFAR10 dataset. The joint use of the residual connection, random interpolation and norm scaling yields the best model performance.
> > >
> > > **How can the authors explain that RandAlign does not improve the performance of GAT (which I suppose oversmooths) in Cora?**
> > >
> > >  *Response:* We have further experimented on the Cora dataset using GAT as the base model. The base model uses Dropout as a regularization method. We found that the joint use of Dropout and RandAlign results in reduced classification accuracy. Therefore, we removed the Dropout method in the base model. Additionally, we found that randomly aligning features with those generated by the first GAT layer yields better results. The results on the Cora dataset are reported in Table 12 (see Page 19). We see that when the number of layers is increased to 16, the model results in significantly reduced performance. With our RandAlign method, the model performance remains almost unchanged with increased layers. The results suggest that our methods helps to alleviate the over-smoothing problem when deeper layers are used.

---

### Decision · Action_Editor_PJQy · 2024-07-22

**Recommendation:** Reject

**Comment:**

Thank you for submitting your manuscript to TMLR. After careful consideration of the reviewers' comments and your detailed responses, we regret to inform you that we have decided not to accept your paper for publication at this time.

The primary concerns raised by the reviewers center around the effectiveness and justification of your proposed method. Specifically, the arguments relating to the norm-based over-smoothing problem need further clarification in relation to the eigenspace-based problem. Additionally, a quantitative comparison demonstrating how RandAlign does not influence the norms of embeddings, unlike SSFG, is necessary to support your motivation.

The reviewers appreciated your active engagement in responding to the reviews. However, they found that the results presented are not sufficiently convincing and require further justification. It remains unclear whether your proposed method is effective and deserving of attention from the broader community.

Several specific issues were noted:

1. Some results that are not the best are still highlighted in boldface, and many results are not significantly better, differing from the best result only within the range of standard deviation error.

2. In the new Table 8, many results for benchmark models are the same without reporting the standard deviation, undermining the credibility of these results.

3. The authors claim that rescaling reduces smoothness lacks a clear theoretical explanation. The reviewers did not observe the oversmoothing issues you claim to address, seeing instead only an improvement in generalizability.

4. Concerns were raised about whether the alleviation of oversmoothing is due to your proposed method or merely a result of the usage of residual connections. The datasets used in your paper are not commonly found in the literature and appear to be highly saturated in terms of accuracy.

5. The results concerning norms and cosine similarity are not convincing. Models with RandAlign show an average similarity higher than 0.95, very close to the similarity observed in models without RandAlign.

6. Regarding the Cora dataset and the GAT model, aligning features with the output of the first layer to maintain high performance suggests that subsequent layers suffer from oversmoothing, even with the inclusion of RandAlign. Modifying the proposed RandAlign equation to make the model work in the Cora and GAT setup indicates that the proposed method is rather weak overall and cannot effectively alleviate oversmoothing.


In conclusion, further theoretical and experimental examination of your proposed method is necessary. The reviewers believe that RandAlign alone is not capable of reducing oversmoothing and certainly not capable of fully alleviating this problem.

We encourage you to address these concerns and consider resubmitting your manuscript after substantial revisions. Thank you for considering TMLR for your work.

Sincerely,

Moshe

**Audience:**

All reviewers, and myself, agree that there is audience for this submission.

**Claims And Evidence:**

After discussions with the authors, the reviewers were not convinced of the results in the paper, and in particular they were not convinced that the proposed method RandAlign can alleviate oversmoothing. In my main comment, I provide more details for the authors.

**Resubmission Of Major Revision:**

The authors may consider submitting a major revision at a later time.